# Formal Data Structures for Tabular Formats in Language Technology

Christian Chiarcos, Max Ionov, Luis Glaser, and Christian Fäth

Applied Computational Linguistics
Goethe University Frankfurt, Germany
`chiarcos|ionov|lglaser|faeth@informatik.uni-frankfurt.de`

**Abstract.** In language technology and the language sciences, tabular formats with tab-separated values (TSV) represent a frequently used formalism to represent linguistically annotated natural language, often addressed as "CoNLL formats". To facilitate interoperability between them, the CoNLL-RDF ontology provides a machine-readable description of 24 such formats that can be used for the conversion between the respective TSV formats and the automated assessment of conversion quality and gaps.

**Keywords:** language technology · data models · CoNLL-RDF · ontology

## 1 Motivation: Incompatible TSV formats

The analysis of natural language requires different, and often, complex steps of processing, traditionally organized in a pipeline architecture. For each of these processing steps, numerous implementations and training data are available. The formats they are based on are different, but often follow a set of common conventions, e.g., those of the Conference of Natural Language Learning (CoNLL, `https://www.conll.org/`). The listing below is an excerpt from the CoNLL 2005 Shared Task,[1] with the first column providing the value of a word (`WORDS`), the second named entity annotations (`NE`), then parts-of-speech (`POS`), shallow constituent syntax (`CHUNK`), clause boundaries (`CLAUSE`), and syntactic parses (`PARSE`).

```
# CoNLL-2005 format                                 | CoNLL-2000 format
#  WORDS          NE   POS   CHUNK   CLAUSE PARSE    | WORD        POS CHUNK
   The            *    DT    (NP*    (S*    (S (NP * | The         DT  B-NP
   spacecraft     *    NN      *)    *             *) | spacecraft  NN  I-NP
   faces          *    VBZ   (VP*)   *      (VP *   | faces       VBZ B-VP
   a              *    DT    (NP*    *      (NP *   | a           DT  B-NP
   ...                                              | ...
```

Much information overlaps with other formats, e.g., CoNLL-2000, but here, the *WORDS* column is labelled *WORD*, the *POS* column is second (not third), and

---

[1] See `https://www.conll.org/previous-tasks` for pointers to the respective websites from which documentation and/or data for most CoNLL formats can be retrieved.

the *CHUNK* column uses IOB encoding scheme[2] (CoNLL-2005 used brackets). Unfortunately, these formats are not all well-documented, but information about them has to be recovered from published papers or directly from data, often leading to confusion about whether a particular tool supports a particular format if it claims to support 'CoNLL'.[3]

## 2   CoNLL-RDF

A problem of CoNLL data is that order, definitions and labels of columns are specific to certain sub-formats, and that other CoNLL formats may provide the same information in different order, with different column labels, or with a different encoding – but that such differences are generally poorly documented, or at least, not at a central place.

CoNLL-RDF [2] is a set of tools for processing and transforming CoNLL and other TSV formats by abstracting from such issues and representing them as an RDF graph. It is similar in scope, function and performance to TARQL,[4] a general-purpose tool for the rendering of CSV data streams as RDF graphs, but it contains a number of optimizations specific to CoNLL data. Most notably, this includes improved capabilities for context-aware stream processing: In CoNLL formats, individual sentences are separated from each other by an empty line, and CoNLL-RDF reads sentence-by-sentence rather than line-by-line. It then allows to apply SPARQL transformations on individual sentences (and their respective context), as well as running queries and serialization in CoNLL formats. The CoNLL-RDF library has been developed as part of the Flexible Integrated Transformation and Annotation Engineering (FINTAN) platform [3], and uses its parallelization of SPARQL transformations to improve scalability and performance. A difference to general-purpose technology for mapping tabular data to RDF (e.g., CSV2RDF or R2RLM), is that CoNLL-RDF uses *linguistic* data structures: The data model uses the NLP Interchange Format [4, NIF] to encode words (`nif:Word`, corresponding to a row), sentences (`nif:Sentence`, group of rows not interrupted by an empty line), and relations between these (`nif:nextSentence`, `nif:nextWord`), and POWLA [1] (`powla:Node`) for phrases. Columns, however, are mapped to properties in the `conll` namespace. This allows to abstract from specifics of the serialization, however, these properties are created on the fly from user-provided labels and have previously not been defined.

---

[2] I, O or B corresponding to each token, B means that a token is the beginning of a chunk, I for a continuation and O for tokens outside of any chunk

[3] See    http://liste.sslmit.unibo.it/pipermail/cwb/2021-March/003980.html for    a    discussion    of    the    CoNLL    support    of    the    Corpus    Workbench (http://cwb.sourceforge.net/). Similarly, the NLTK CoNLL Corpus Reader (https://www.nltk.org) does not seem to support the highly popular CoNLL-U and CoNLL-X formats – although this is not evident from its documentation.

[4] https://tarql.github.io/

## 3   CoNLL-RDF ontology

The CoNLL-RDF ontology, introduced with this paper, adds machine-readable semantics for existing datasets encoded as CoNLL-RDF and provides the basis for linking RDF corpora with other Semantic Web resources. As it provides the first systematic documentation of CoNLL dialects, we expect the CoNLL-RDF ontology also to have more general applications for the automated processing of CoNLL files in NLP.

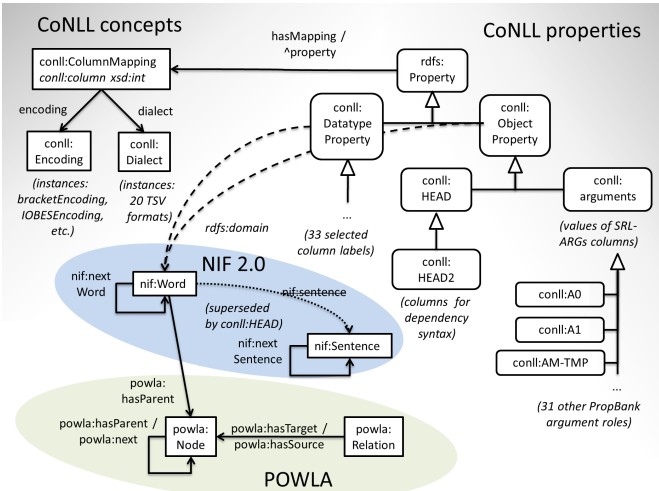

**Fig. 1.** CoNLL-RDF Ontology: Classes and properties of CoNLL and external vocabularies

It consists of two components: (1) classes, properties and axioms used to define formats (Fig. 1), and (2) the machine-readable description of existing CoNLL and related formats. Aside from documenting the use of the NIF and POWLA vocabularies, the contribution of the ontology is two-fold:

- Annotation properties in the `conll` namespace, along with human-readable definitions and labels: 33 datatype properties that reflect previously used column labels, 2 object properties corresponding to columns for dependency syntax (`HEAD`, `HEAD2`) and 34 object properties for semantic roles. To model domain restrictions over these properties, we introduce `conll:Datatype-Property` and `conll:ObjectProperty` as superproperties.
- Concepts and properties that describe the mapping from CoNLL properties to different formats (dialects) and the respective encoding preferences. A column mapping links a CoNLL property (`conll:property`) with a particular column position (`conll:column`) in a particular format (`conll:dialect`). Any CoNLL property can be related to multiple mappings. Each relation

then describes a mapping for a specific property in a specific dialect. This allows to represent data independent of the exact dialect. Moreover, the column mapping can define the encoding strategy of phrases and multi-word expressions by means of the `conll:encoding` property, e.g., as using bracket or IOB encoding. Note that the same property can be encoded in different ways, as shown for the bracket notation and the IOB encoding above.

With this, we can describe how to retrieve equivalent CoNLL-RDF data from different formats, e.g., for the `CHUNK` column of CoNLL-2000:

```
:CoNLL-00 a :Dialect;
  rdfs:label "CoNLL-00 format";
  rdfs:isDefinedBy <https://www.clips.uantwerpen.be/conll2000/chunking/>.
:CHUNK rdfs:subPropertyOf :DatatypeProperty ; rdfs:label "CHUNK";
  rdfs:comment "The chunk tags contain the name of the chunk type, for example I-NP ..."@en;
  :hasMapping [ a :ColumnMapping; :column "4"; :dialect :CoNLL-05; :encoding :bracketEncoding ] ;
  :hasMapping [ a :ColumnMapping; :column "3"; :dialect :CoNLL-00, :encoding :iobEncoding ] . # etc
```

## 4   Outlook: Conversion and convertibility

The CoNLL-RDF ontology provides machine-readable semantics for an inventory of CoNLL properties (and classes) for a growing collection of two dozen CoNLL and related formats currently used in language technology. With the ontology, the relations between these formats have been made explicit, so that it now becomes possible to (a) automatically transform one TSV format into another, resp. (b) to assert/infer that a particular format cannot be automatically transformed into another. We illustrate this by CoNLL-Transform,[5] a command-line tool that uses the CoNLL-RDF ontology to generate converters and compatibility analyses for 552 combinations of input and output formats. The ontology, CoNLL-RDF and CoNLL-Transform are available as open source from `https://github.com/acoli-repo/conll`. The ontology is archived at Zenodo[6] and resides under `http://purl.org/acoli/conll#`. The recommended namespace prefix is `conll:`.

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
