# OpenReview forum: "Formal Data Structures for Tabular Formats in Language Technology"
_eswc-conferences.org/ESWC/2021/Conference/Poster_and_Demo_Track — Submitted to ESWC2021 P&D_

### Official Review · ~Yoan_Chabot1 · 2021-04-13
**A useful ontology for the NLP community completing a rich toolbox**

**Rating:** 9
**Confidence:** 3

**Review:**

In this paper, the authors introduce the CoNLL-RDF ontology providing a machine-readable format for 24 heterogeneous CoNLL TSV formats.
The paper is very clear and well written.
The organization of the paper allows the readers to understand the importance of the problem, to know the context and the tools already proposed by the authors and to position the contribution (CoNLL-RDF ontology) in relation to this context.

The contribution is interesting and useful for the community and completes an already rich toolbox.
The ontology, extremely well documented, is available on https://zenodo.org/record/4361476 (footnote 6 https://zenodo.org/record/4361476/ not working) and all the tools around it are available on Github.
As mentioned by the authors, this contribution paves the way for automatic conversion tools between these formats as well as for other automatic applications in the NLP domain.

Section 1 presents the problem very clearly, even for readers outside the NLP community.
This section presents well the motivations and issues by highlighting the heterogeneity of CoNLL formats between different editions of the conference.
The example provided in the figure illustrates the problem well with a focus on the 2005 and 2000 editions of the CoNLL format.
Conversion from one format to another requires a good knowledge of the formats.
However, the documentation of the different formats is insufficient according to the authors and the information has to be collected from disparate sources.
The modeling effort proposed in the paper is therefore beneficial for the NLP community and the proposed tools will undoubtedly be very useful and used.

The ontology and the proposed tools support 24 CoNLL formats. I think this corresponds to the different editions (https://www.conll.org/previous-tasks) + CoNLL-U?
A pointer to the list of supported formats could be added in the paper.
This precision on the number of supported formats suggests that the toolbox does not support all CoNLL formats.
If this is the case, what are the reasons for this limited support?

To improve the flow of the paper, I suggest that the authors merge the first paragraph of section 2 with the end of section 1 as the arguments are repeated between these two paragraphs.

The following section presents the specificities of the CoNLL-RDF toolkit (introduced in a previous paper by the same authors) compared to more generic tools already existing like TARQL and CSV2RDF.
The section is well argued and provides the readers with enough context to fully understand the contribution of this paper: the CoNLL-RDF ontology.

Figure 1, although a bit cluttered, gives a good overview of the organization of the ontology.
The ontology is based on two existing semantic models well referenced in the paper: NIF and POWLA.
The authors then present the properties and concepts introduced in the conll namespace in a rather quick way, but consulting the TTL file and Figure 1 in parallel allows the reader to understand the modeling.
By defining the dialects and the mappings between these dialects (bravo for the modeling effort), the CoNLL-RDF ontology nicely completes the CoNLL-RDF toolbox.
In conclusion, the authors illustrate the interest of the ontology by proposing the CoNLL-Transform tool.

**Anonymity:**

No, I would like my review to be deanonymized.

---

### Official Review · ~Vassil_Momtchev1 · 2021-04-14
**Unclear contributions compared to a previous paper and lack of any form of evaluation/discussion/result assessment**

**Rating:** 5
**Confidence:** 4

**Review:**

A very similar paper "CoNLL-RDF: Linked Corpora Done in an NLP-Friendly Way (May 2017)" (DOI: 10.1007/978-3-319-59888-8_6) has been published @ International Conference on Language, Data and Knowledge. I find it quite hard to differentiate what's the new contribution here beyond the Github project.

The paper presents the challenge of the different serialization formats on ConLL. It introduces an RDF-based schema to generalize the data model and a Github project for bi-directional transformation between the various formats.

Pros:

* It presents a clear contribution for avoiding writing transformation scripts between 552 theoretical combinations
* It looks that the referenced Github project has far more features than the one listed in the paper, although this puzzles me if I get the full picture
* The CoNLL-RDF schema is resolvable -

Cons:

* I miss what is it the practical application of the submitted work - by design CoNLL formats are optimized for different challenges; presenting any motivation or case when these tools are needed will be of huge benefit for the readers
* It's unclear if/how the generic data model to help to achieve any semantic interoperability on the value level
* I miss any form of evaluation/discussion if all transformations meaningful and if there will any loss of information

A quick check of the ontology puzzled me why:

:DatatypeProperty rdf:type owl:DatatypeProperty ,
                       owl:FunctionalProperty ;

This seems like an unnecessary test for any reasoners to see if it can handle "1" owl:sameAs "2" without breaking.

**Anonymity:**

No, I would like my review to be deanonymized.

---

### Official Review · AnonReviewer3 · 2021-04-14
**Unclear novelty with respect to previous publications**

**Rating:** 5
**Confidence:** 4

**Review:**

The paper presents an approach to harmonize tabular formats used for language technology using RDF.
I could not understand if the paper presents a demo or a poster, but I am leaning towards the second interpretation (in the first case the paper fails to explain the demo adequately).

PROS

Supporting testing of language technology is an important task and relevant for the ESWC community.

The approach described in the paper seems rather mature, as the only few statistics reported suggest (to generate converters and compatibility analyses for 552 combinations of input and output formats.)

CONS

My main concern is about the precise definition and novelty of the proposed solution.

Definition and novelty. It is not clear what is the focus of this paper: the CoNLL-RDF ontology? But the CoNLL-RDF ontology refers to a 2017 publication.

The conversion tools in https://github.com/acoli-repo/conll ? But the latter is only marginally described in the paper.


SUMMARY

The work described in this paper is relevant and useful, but it is hard to distill the actual contribution of this paper from related publications of the same authors. If the novelty is in the tools made available in GitHub the paper should focus on these ones and refer to previous work for the ontology (e.g., remove Figure 1 that is hard to read and occupy much space). If the paper describes a demo, a more clear description of what will be demoed is needed. In both cases, a clear specification of the contributions of this specific paper/demo is needed.


**Anonymity:**

Yes, I would like my review to remain anonymous.

---

### Official Review · AnonReviewer1 · 2021-04-15
**Well written but the novelty is not clear**

**Rating:** 5
**Confidence:** 2

**Review:**

The paper introduces a CoNLL-RDF ontology for facilitating interoperability between different formats of natural language resources.

- The paper is very clearly written.
- The converters are available online.

However, my question would be that CoNLL2RDF has already been proposed before as the authors point to the reference. The novelty of the paper seems to be the definition of the ontology on top of the output of CoNLL2RDF. This seems to be very limited. The defined ontology seems to have very few number classes. It will be good to have more detailed statistics.

**Anonymity:**

Yes, I would like my review to remain anonymous.

---

### Official Review · Program_Chairs · 2021-04-18
**Metareview: Reject (Unclear added contribution versus previous work)**

**Rating:** 5
**Confidence:** 5

**Review:**

Though the reviewers acknowledge the usefulness of the work, the quality of the documentation, etc., three of the reviewers express concerns that it is not clear what is the added contribution or value added by this paper in comparison to the 2017 publication.

For this reason, we cannot accept this paper at this time.

**Anonymity:**

Yes, I would like my review to remain anonymous.

---

### Decision · Program_Chairs · 2021-04-19

Reject